

# Role of in situ-excited planetary waves in polar vortex splitting during the 2002 Southern Hemisphere sudden stratospheric warming event

Ji-Hee Yoo[1] and Hye-Yeong Chun[1]

[1]Department of Atmospheric Sciences, Yonsei University, Seoul, 03722, South Korea

*Correspondence to*: Hye-Yeong Chun (chunhy@yonsei.ac.kr)

**Abstract.** On 25 September 2002, the Southern Hemisphere experienced its first and only major sudden stratospheric warming (SSW02) since routine upper-atmosphere observations commenced in 1957. The sudden splitting of the polar vortex, a phenomenon rarely observed even in the Northern Hemisphere, marked this event. While previous studies focused on tropospheric waves and vortex preconditioning, the role of in situ-excited planetary waves (PWs) remains unexplored. The current study addresses this gap by examining the impact of in situ-generated PWs on SSW02 evolution. As the onset approached, the displaced polar vortex elongated and ultimately split into two vortices. The explosive amplification of zonal wavenumber (ZWN) 2 PWs (PW2) at 10 hPa, which split the vortex, was not only driven by upward-propagating PW2 from the lower stratosphere but also by westward-propagating PW2 excited in situ in the mid-to-upper stratosphere, which then descended to 10 hPa. This spontaneous PW2 generation was associated with barotropic–baroclinic instability, triggered as the stratosphere became dominated by easterlies descending from the lower mesosphere. The unusual poleward shift of the polar vortex facilitated easterly development by directing ZWN1 PWs (PW1) into the polar stratosphere, where they deposited strong westward momentum. PW2 amplification via instability occurred through two mechanisms: (1) the breaking of PW1 generated smaller-scale waves through energy cascading while inducing instability that amplified these smaller-scale waves, which could play a role in the local PW2 growth; and (2) over-reflection of upward-propagating PW2. While both mechanisms contributed to the amplification, the latter became dominant as the onset neared.

## 1 Introduction

On 25 September 2002, the first major sudden stratospheric warming (SSW) event (hereafter referred to as SSW02) was recorded in the Southern Hemisphere (SH), marking an unprecedented occurrence in Antarctic stratospheric observations since 1957. Although minor midwinter warmings have occasionally been observed in the SH, a major warming event characterized by the complete breakdown of the polar vortex during midwinter has only been documented in this instance. The rarity of SSW in the SH has been attributed to several factors, including the sparse mountainous terrain; weak longitudinal land–sea contrasts; and the nearly zonally symmetric, cold, elevated Antarctic surface (Gray et al., 2005). These factors collectively suppress SSW in the SH by weakening planetary wave (PW) forcing and strengthening the polar-night



jet (PNJ). However, SSW02 was distinguished by a record-breaking weakening of the PNJ and extremely high temperatures,
both of which remain unparalleled in the SH climate.

Notably, SSW02, the first observed major SSW event in the SH, was of the split type, a phenomenon far less common than
displacement-type warmings in the Northern Hemisphere (NH, Charlton et al., 2005). The vortex split significantly impacted
the typically quiescent Antarctic ozone hole, causing it to divide into two separate regions (Allen et al., 2003; Baldwin,
2003). Using a mechanistic model, Manney et al. (2005) demonstrated that the vortex split could be simulated exclusively
with zonal wavenumber (ZWN) 2 waves at the 100 hPa pressure level without requiring vortex preconditioning. This finding
suggests that anomalously strong wave forcing in the lower stratosphere was the primary driver of the major warming.
Krüger et al. (2005) attributed the intense stratospheric wave forcing to unusually strong tropospheric wave pulses,
collectively formed by the quasi-stationary PWs of ZWNs 1–3. However, unlike Manney et al. (2005), they emphasized that
the substantial weakening of the PNJ during early winter (July–August 2002) served as a crucial preconditioning factor that
significantly amplified the upward propagation of tropospheric waves into the stratosphere (see also Baldwin, 2003).

Newman and Nash (2005) observed that the PNJ was preconditioned to not only weaken significantly but also to shift
unusually poleward, thereby directing tropospheric waves toward the pole across a broad altitude range into the middle
stratosphere. This abnormally poleward-shifted vortex structure was accompanied by the westerly quasi-biennial oscillation
(QBO) in the lower stratosphere (30–50 hPa) and anomalous easterlies in the equatorial upper stratosphere (1–10 hPa),
where the semiannual oscillation (SAO) dominates (Gray et al., 2005; Newman and Nash, 2005). Given that the westerly
QBO is generally unfavorable for initiating SSWs (Holton and Tan, 1982) and that SSWs typically begin with wind reversal
in the upper stratosphere and lower mesosphere, Gray et al. (2005) examined the influence of anomalous equatorial easterlies
in the upper stratosphere on the onset of SSW02. Their analysis revealed that this wind structure could have contributed to
triggering SSW02 by confining equatorward-propagating PWs to the pole during early and midwinter. Meanwhile, Charlton
et al. (2005) emphasized that the vortex split dynamics involved complex nonlinear interactions within the coupled
troposphere–stratosphere system, suggesting that the conventional framework of lower atmospheric forcing may be
insufficient to comprehensively account for this event.

While previous research on SSW02 primarily focused on the role of tropospheric waves and vortex preconditioning in
directing these waves toward the polar stratosphere, recent studies on SSWs in the NH have increasingly highlighted the role
of in situ-excited PWs within the stratosphere or mesosphere. Regarding the split SSW event in January 2009, Song et al.
(2020) suggested that the eastward-propagating PWs of ZWN2 (PW2) in the lower mesosphere partially contributed to
vortex splitting at 10 hPa through downward propagation. With regard to the mechanisms driving the eastward-traveling
PW2, Iida et al. (2014) proposed wind shear instability as a localized source, whereas Rhodes et al. (2021) attributed it to the
over-reflection of upward-propagating tropospheric PW2. Yoo et al. (2023, hereafter YCK23) demonstrated that unstable





PW2, spontaneously generated within the stratosphere, played a significant role in vortex splitting during the SSW event in
2021 (hereafter, SSW21).

This study examines whether in situ-excited PWs contributed to the polar vortex split during the first and only SSW event in
the SH, as observed in several NH SSWs. Specifically, this research identifies the spontaneous excitation of PWs in the mid-
to-upper stratosphere and their subsequent downward propagation toward 10 hPa—a feature overlooked in previous studies
on SSW02, which focused on altitudes at or below 10 hPa. Thus, building on the approach adopted in YCK23, this study
seeks to determine the in situ source of these waves and clarify the underlying preconditioning mechanism. In this context,
we draw comparison with SSW21, the focus of YCK23, to examine similarities and differences in the governing dynamics.
As we are aware, this study is the first to investigate the role of locally generated PWs in the development of SSW02,
offering deeper insight into the processes driving its occurrence.
**2 Data and analysis methods**
**2.1 Modern-Era Retrospective analysis for Research and Applications, version 2 (MERRA-2) data**
This study utilizes MERRA-2 reanalysis data, which exhibits a horizontal resolution of $0.625° \times 0.5°$ (longitude $\times$ latitude)
and a 3 h temporal resolution, covering altitudes from the surface to 0.1 hPa (Gelaro et al., 2017). The dataset spans a 44-
year period (1980–2023). All results presented are based on daily averages.
**2.2 Analysis methods**
**2.2.1 Eliassen–Palm flux (EP-flux) and its divergence**
The EP-flux and its divergence (EPFD), which represent wave activity flux and wave forcing, respectively, are calculated as
follows (Andrews et al., 1987):
$$\boldsymbol{F} = \left(F^\phi, F^z\right) = \rho_0 a \cos\phi \left(-\overline{u'v'} + \bar{u}_z \frac{\overline{v'\theta'}}{\bar{\theta}_z}, \left[f - \frac{1}{a\cos\phi}(\bar{u}\cos\phi)_\phi\right] \frac{\overline{v'\theta'}}{\bar{\theta}_z} - \overline{u'w'}\right), \qquad (1)$$

$$\boldsymbol{\nabla} \cdot \boldsymbol{F} = \frac{1}{a\cos\phi} \frac{\partial}{\partial\phi}\left(F^\phi \cos\phi\right) + \frac{\partial F^z}{\partial z}. \qquad (2)$$

In the above equations, $\phi$ and $z$ denote latitude and log-pressure height, respectively. $\rho_0$ represents the reference density, $a$
denotes Earth's mean radius, and $f$ denotes the Coriolis parameter. The parameters $u$, $v$, and $w$ correspond to the zonal,
meridional, and vertical wind components, respectively, while $\theta$ denotes potential temperature. The overbar $\overline{()}$ and prime (')
denote the zonal mean and deviations from the zonal mean, respectively. The EP-flux vector, denoted as $\boldsymbol{F}$, consists of
meridional ($F^\phi$) and vertical ($F^z$) components. EPFD is defined as $(1/\rho_0 a \cos\phi)\boldsymbol{\nabla} \cdot \boldsymbol{F}$.



### 2.2.2 Barotropic (BT)–baroclinic (BC) instability

The evaluation of BT–BC instability is based on the meridional gradient of the quasi-geostrophic potential vorticity (QGPV, Andrews et al., 1987):

$$\bar{q}_y = \beta - \bar{u}_{yy} - \frac{1}{\rho_0}\left(\rho_0 \frac{f^2}{N^2}\bar{u}_z\right)_z, \tag{3}$$

where $\bar{q}$, $\beta$, and $N$ denote the zonal-mean QGPV, meridional derivative of $f$, and Brunt–Väisälä frequency, respectively. The necessary condition for BT–BC instability is met when the typically positive $\bar{q}_y$, associated with wintertime circulation, turns negative (Salby, 1996). In Sect. 3, we collectively define the first two terms on the right-hand side as the barotropic term, while the third term is designated as the baroclinic term.

### 2.2.3 Linearized disturbance QGPV equation

In log-pressure coordinates, the linearized disturbance QGPV equation is expressed as follows (Matsuno, 1970; 1971):

$$\left(\frac{\partial}{\partial t} + \bar{u}\frac{\partial}{a\cos\phi\,\partial\lambda}\right)q' + v'\frac{\partial\bar{q}}{a\partial\phi} = \frac{1}{a\cos\phi}\left[\frac{\partial Y'}{\partial\lambda} - \frac{\partial(X'\cos\phi)}{\partial\phi}\right] + \frac{f_0}{\rho_0}\frac{\partial}{\partial z}\left[\rho_0 \frac{Q'}{e^{\frac{\kappa}{H}z}\left(\frac{\partial\bar{T}_0}{\partial z} + \frac{\kappa\bar{T}_0}{H}\right)}\right], \tag{4}$$

$$q' \equiv \frac{1}{fa^2}\left[\frac{1}{\cos^2\phi}\frac{\partial^2\Phi'}{\partial\lambda^2} + \frac{f^2}{\cos\phi}\frac{\partial}{\partial\phi}\left(\frac{\cos\phi}{f^2}\frac{\partial\Phi'}{\partial\phi}\right) + \frac{f^2 a^2}{\rho_0}\frac{\partial}{\partial z}\left(\frac{\rho_0}{N^2}\frac{\partial\Phi'}{\partial z}\right)\right], \tag{5}$$

$$\frac{\partial\bar{q}}{a\partial\phi} \equiv \frac{2\Omega\cos\phi}{a} - \frac{1}{a^2}\frac{\partial}{\partial\phi}\left[\frac{1}{\cos\phi}\frac{\partial(\bar{u}\cos\phi)}{\partial\phi}\right] - \frac{1}{\rho_0}\frac{\partial}{\partial z}\left(\rho_0\frac{f_0^2}{N^2}\frac{\partial\bar{u}}{\partial z}\right). \tag{6}$$

In the above equations, $\lambda$ denotes longitude, and $q'$ represents the QGPV perturbation. The perturbations of the zonal and meridional components of gravity wave drag (GWD) from their zonal mean are represented by $X'$ and $Y'$, respectively. $Q'$ denotes the perturbation diabatic heating rate, while $\psi'$ represents the perturbation streamfunction (defined as $\psi' = \phi'/f_0$, where $\phi'$ denotes the perturbation geopotential). On the right-hand side of Eq. (4), the first bracketed term represents the nonconservative forcing term of the QGPV perturbation associated with GWD (Song et al., 2020). We investigate whether the nonconservative GWD forcing, defined as $Z'$ below, contributed to the vortex split using the zonal and meridional components of the parameterized GWD data (McFarlane, 1987; Molod et al., 2015).

$$Z' = \frac{1}{a\cos\phi}\left[\frac{\partial Y'}{\partial\lambda} - \frac{\partial(X'\cos\phi)}{\partial\phi}\right] \tag{7}$$

### 2.2.4 Squared refractive index

To investigate PW propagation, we use the squared refractive index, defined as (Andrews et al., 1987)



$$n^2 = \left[\frac{\bar{q}_\phi}{a(\bar{u} - C_x)} - \left(\frac{k}{a\cos\phi}\right)^2 - \left(\frac{f}{2NH}\right)^2\right] a^2. \tag{8}$$

where $k$ denotes the ZWN, and $C_x$ represents the zonal phase speed of the wave. PWs can propagate in regions where $n^2$ is
positive, whereas their propagation is impeded in regions where $n^2$ is small or negative (Karoly and Hoskins, 1982).
**3 Results**
**3.1 Variations in wind and temperature during SSW02**
Figure 1 presents the temporal evolution of the zonal-mean zonal wind at 60°S and the polar cap temperature over 60–90°S
during SSW02 development. The reversal of zonal-mean westerlies to easterlies began in the lower mesosphere on 22
September and propagated downward to 10 hPa within three days, marking the onset of major SSW02 (Charlton and Polvani,
2007). Over the following week (18–25 September), the PNJ weakened dramatically by more than 100 m/s, accompanied by
a sudden temperature increase of approximately 20 K at 10 hPa. The deceleration and eventual reversal of westerlies and the
temperature rise were both statistically significant at the 99% confidence level. A key difference between SSW02 and
SSW21 is that in SSW02, the pronounced weakening and reversal of zonal wind and the corresponding temperature increase
manifested as an upward-propagating signal from the troposphere. This observation is consistent with that of Krüger et al.
(2005), who identified a tropospheric zonal wind pattern extending upward into the stratosphere from 16 September 2002.
**3.2 Dynamical features of the vortex split and associated PW activities**
Figure 2a offers an overview of the polar vortex split during the onset of SSW02 by illustrating the geopotential height
perturbation (GHP) and horizontal wind fields at 10 hPa. By 21 September, the vortex had weakened and shifted away from
the pole, and from 23 September, it elongated and ultimately split into two distinct vortices. The associated PW activities are
analyzed by examining the evolution of ZWN1 and ZWN2 PW amplitudes at 60°S (Fig. 2b). In the days leading up to the
onset, ZWN1 PW (PW1) had a greater amplitude than PW2 until 23 September, after which it rapidly weakened. Conversely,
PW2 intensified and surpassed the amplitude of PW1 at 10 hPa for two days following the onset. This anticorrelation
between PW1 and PW2 is a common feature of split-type SSW events, including SSW21. However, PW activity during
SSW02 differed markedly from that during SSW21. Specifically, during SSW02, PW2 was enhanced from the upper
troposphere—similar to the zonal wind and temperature anomalies (Fig. 1)—and exhibited statistically significant positive
anomalies. In contrast, during SSW21, PW2 strengthened within the mid-stratosphere (Fig. 1b of YCK23). This suggests
that PW2, originating from the troposphere, played a crucial role in the vortex split during SSW02, aligning with previous
findings (Krüger et al., 2005; Manney et al., 2005).



However, further insight is provided by the GHP of PW2 in the longitude–height cross sections depicted in Fig. 2c, which
reveals an eastward tilt of the phase with increasing altitude above 10 hPa during the PW2 amplification period (21–25
September). This suggests downward-propagating PW2 from the mid-to-upper stratosphere, potentially contributing to PW2
intensification at 10 hPa.





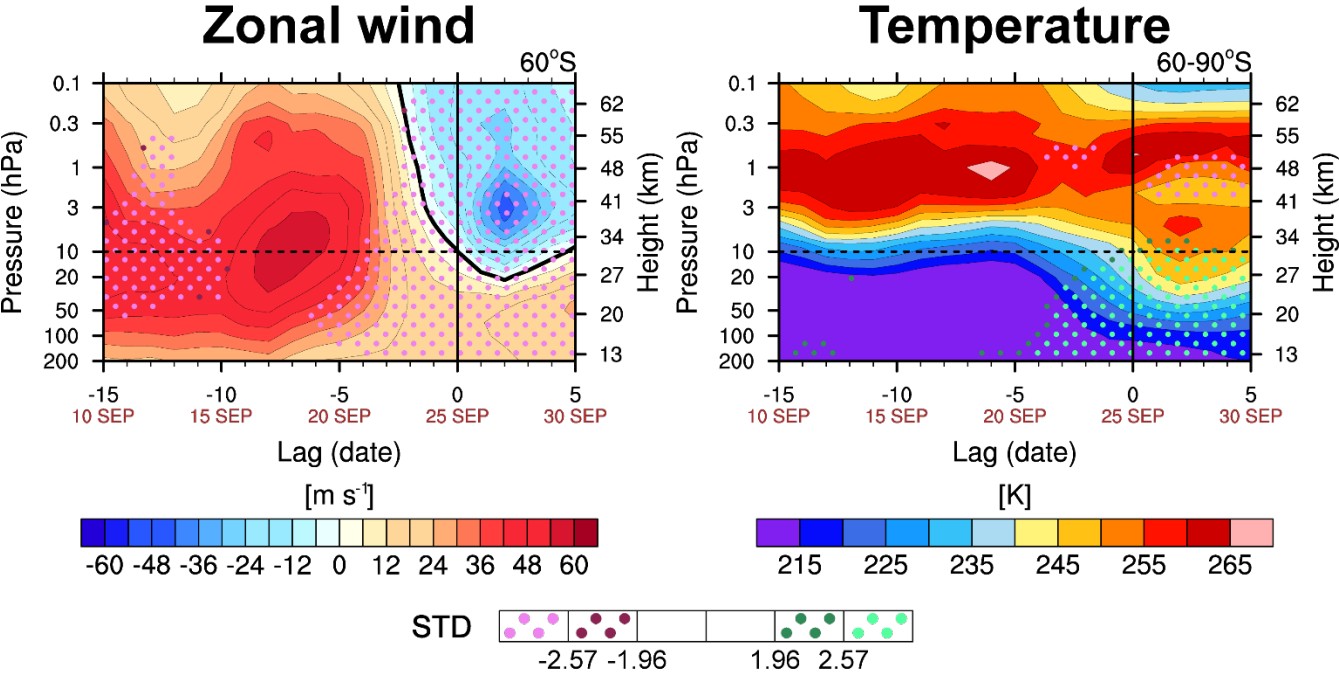


**Figure 1:** Time–height cross sections of the zonal-mean zonal wind at 60°S (left) and polar cap temperature averaged over 60–90°S (right). Dark and bright pink (green) dots indicate regions where the analyzed variable is algebraically smaller (larger) than its 44-year climatology by more than 1.96 and 2.57 standard deviations (STD), signifying statistical anomalies at the 95% and 99% confidence levels, respectively.







**Figure 2:** (a) Polar stereographic series showing horizontal wind speed (shading) and GHP from the zonal mean (contours) at 10 hPa on 21, 23, and 25 September 2002. Red (blue) contours denote positive (negative) values. (b) GHP amplitude of PWs with ZWN1 (PW1, left) and 2 (PW2, right) at 60°S. (c) Longitude–height cross sections of PW2 GHP on 21, 23, and 25 September 2002.



### 3.3 In situ source of the downward propagating PW2 in the mid-to-upper stratosphere

The downward-propagating signal of stratospheric PW2 is more evident in the EP-flux and EPFD, as depicted in Fig. 3a. Beginning on 22 September, downward EP-fluxes emerged from the region of positive EPFD (70–50°S above 10 hPa), which was located within the easterlies evolving from the polar mesosphere. As the easterlies intensified and extended toward the equator, the positive EPFD also strengthened, exhibiting statistically significant positive anomalies that exceeded the 95% confidence level. This indicated in situ PW2 excitation within the stratosphere and its dependence on the background atmospheric conditions. The downward- and equatorward-propagating stratospheric PW2 converged with upward-propagating tropospheric PW2 near 10 hPa and 60°S, leading to a significant negative EPFD at the 99% confidence level.

The wave fluxes and forcings observed during SSW02 closely resembled those during SSW21 (Fig. 3a of YCK23), where BT–BC instability was the primary source of stratospheric in situ PW2 excitation. Accordingly, we examined the potential role of instability as a source by analyzing $\bar{q}_y$ (Fig. 3b). Negative $\bar{q}_y$ appeared within the easterly region and intensified as it expanded equatorward, mirroring the evolution of easterlies. Recognizing a similar pattern, YCK23 attributed the onset of instability to the strengthening easterlies, as follows: The positive meridional curvature of the easterlies exceeded the beta effect, rendering the barotropic term negative. Simultaneously, the negative shear and positive curvature of the easterlies caused the baroclinic term to turn negative. These factors collectively led to negative $\bar{q}_y$ (see Figs. 3 and 4 in YCK23 for further details). The region of instability largely overlapped with the area of PW2 generation. All these features, consistent with YCK23, indicate that strong shear instabilities driven by strengthening easterlies promoted unstable PW2 growth within the stratosphere during SSW02.

Zonally asymmetric GWD could also generate PWs in situ in the upper stratosphere and lower mesosphere through the nonconservative forcing ($Z'$) of the linearized disturbance QGPV (Eq. 7). Song et al. (2020) reported that the significant PW2 amplification at 10 hPa, which led to the splitting of the polar vortex during the 2009 SSW, was partially attributed to the downward-propagating PW2 generated in situ by ZWN2-patterned GWD in the lower mesosphere. Yoo et al. (2024) revisited this excitation mechanism using an idealized general circulation model and demonstrated that PWs induced by $Z'$ led to substantial fluctuations and forcings as they propagated. To examine whether stratospheric PW2 is associated with GWD via $Z'$, we analyzed the magnitude of ZWN2 $Z'$ along with the divergence of PW2 EP-fluxes (Fig. 3c). Notably, ZWN2 $Z'$ showed a large amplitude primarily in the upper stratosphere and lower mesosphere (0.3–0.1 hPa), where PW2 generation by GWD occurred during the 2009 SSW, as identified by Song et al. (2020). However, these areas did not coincide with the key region of PW2 excitation (5–1 hPa) during SSW02. Even within the lower mesosphere, $Z'$ in the 60–70°S region, where positive EPFD appeared, was weaker than that in other latitudinal regions. Therefore, as in SSW21, instability was identified as the most likely source of PW2 in this case.





**Figure 3:** Latitude–height cross sections of (a) the EP-flux (vectors) overlaid on its divergence (EPFD, shading) for PW2, (b) negative meridional gradient of the zonally averaged quasi-geostrophic potential vorticity ($\bar{q}_y$, colors) overlaid with the positive EPFD of PW2 (red contours), and (c) magnitude of the nonconservative GWD forcing of the quasi-geostrophic potential vorticity perturbation ($Z'$, shading) overlaid with the positive EPFD of PW2 (red contours) from 22 to 25 September 2002. Black contours represent zonal-mean zonal wind, with solid, dashed, and thick solid lines indicating positive, negative, and zero wind, respectively.



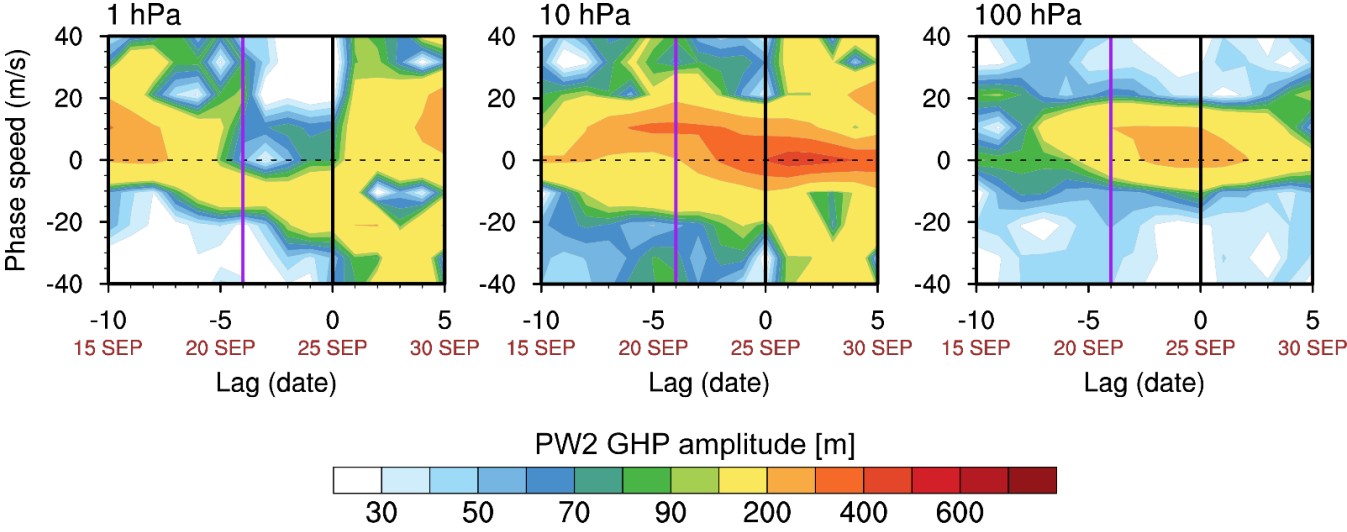

**Figure 4:** Time–zonal phase speed cross sections of PW2 GHP amplitude averaged over 45–75°S at 1, 10, and 100 hPa. The purple and black vertical lines indicate the start date of PW2 amplification and the onset date, respectively.



An inspection of the zonal phase speed of in situ-excited PW2 supports this hypothesis. Instability destabilizes PWs whose
zonal phase speed matches the zonal wind speed in the instability region (Dickinson, 1973). Figure 4 illustrates the time–
zonal phase speed cross sections of the PW2 GHP amplitude at 1, 10, and 100 hPa. During the generation period of mid-to-
upper stratospheric PW2 (22–25 September), PW2 at 1 hPa predominantly exhibited westward phase speeds of up to 30 m
$s^{-1}$, aligning with the range of easterlies present in the instability region (0–30 m $s^{-1}$, Fig. 3b). These waves could not be
solely attributed to the direct upward propagation of PWs from 100 hPa, where eastward-propagating waves were more
prevalent. Collectively, all accumulated evidence suggests that stratospheric westward-propagating PW2 (WPW2) arose
spontaneously from their critical levels within the instability region.

These in situ-excited WPW2 influenced the enhancement of PW2 at 10 hPa through downward propagation. This is
evidenced by the phase speed range of WPW2 across different altitudes. While the phase speed of WPW2 at 100 hPa was
largely below 10 m $s^{-1}$, it increased to above 30 m $s^{-1}$ at 10 hPa, aligning with the phase speed range observed at 1 hPa.
Furthermore, as the onset neared, the increasing trend in the westward phase speed range at 10 hPa closely followed that at 1
hPa, contrasting with the decreasing trend at 100 hPa. These findings confirm that in situ-excited WPW2 at 1 hPa
contributed to amplifying PW2 at 10 hPa. Notably, this contribution persisted even after the onset date.
**3.4 Vortex preconditioning: poleward shift of the polar vortex**
Consistent with SSW21, the evolution of easterlies within the polar stratosphere drove the vortex toward BT–BC instability
during SSW02. This raises the question of whether SSW02 was also preceded by double-westerly jets and their critical-level
interaction with tropospheric PWs, which led to zonal wind reversal and associated instability during SSW21. To address
this, we analyzed the evolution of zonal-mean zonal winds from 20 September to the onset date (Lag = –5 to 0), as shown in
Fig. 5. A double-westerly jet–like configuration appeared on 21 September, with one core in the polar stratosphere and
another in the subtropical mesosphere. However, the equatorial stratospheric easterlies that propagated toward the polar
stratopause along the path between the two cores and eventually dominated the polar stratosphere—a phenomenon observed
in SSW21 (see Fig. 7 of YCK23)—were not identified in SSW02. Instead, easterlies emerged from the polar mesosphere on
22 September (Lag = –3) and rapidly descended into the lower stratosphere.



# Zonal-mean zonal wind

**Figure 5:** Latitude–height cross sections of zonal-mean zonal wind in the SH from 20 to 25 September 2002.





## (a) PW1 EPFy

## (b) PW1 EPF/EPFD

**Figure 6**: Latitude–height cross sections of (a) the meridional component of EP-flux (EPFy, shading) and (b) EP-flux (vectors) overlaid on EPFD (shading) for PW1 from 20 to 25 September 2002. Black contours indicate zonal-mean zonal wind, with the same contour specifications as in Fig. 3.





An abrupt development of easterlies preceded the abnormal increase (decrease) in the zonal-mean zonal wind on the
poleward (equatorial) side of the jet streak on 20–21 September. This is consistent with the poleward shift of the PNJ relative
to climatology, as documented by Newman and Nash (2005), although they noted that this shift had begun as early as April.
Throughout the winter period, this shift guided irregular bursts of tropospheric waves toward higher latitudes, enhancing
westward momentum transfer into the polar region. While Fig. 5 focuses on the period of rapid wind transition, a feature
supporting this argument is evident in the meridional component of EP-flux ($F^\phi$) and EPFD of PWs during this period (Fig.
6). The focus here is on PW1, which predominantly contributed to negative PW forcings (not shown). From 20 to 21
September, $F^\phi$ exhibited significant negative values along the vortex center, reaching the 99% confidence level. This
confirms an unusual progression of PW1 toward higher latitudes, guided by the poleward-displaced vortex. Following the
rapid weakening and transition of the PNJ into easterlies, a substantial negative $F^\phi$ value gradually extended to lower
altitudes by 23 September. These waves deposited statistically significant negative EPFD, exceeding the 95% confidence
level, with a maximum of approximately 50 m s$^{-1}$ day$^{-1}$ near the jet maximum on 21 September (Fig. 6b). Exceptionally
strong westward momentum from PW1 facilitated the transition of westerlies to easterlies from the polar mesosphere.
Positive feedback via critical-level interaction between the zero-wind line and subsequent tropospheric PW1 further
enhanced the downward expansion of polar easterlies with increasing intensity.

The next key question concerns the mechanisms driving the unusual poleward displacement of the vortex. Newman and
Nash (2005) and Gray et al. (2005) identified anomalous easterlies in the equatorial upper stratosphere (1–10 hPa) as a
potential factor contributing to the poleward vortex shift. Gray et al. (2005) proposed that these upper stratospheric easterlies
over the equator generated strong horizontal wind shear and steep PV gradients in the SH subtropics during early winter,
effectively confining equatorward-propagating PWs to the polar upper stratosphere. In this context, they suggested that the
tropical stratopause's SAO as a favorable precursor to SSW.

### 3.5 Destabilization of ZWN2 waves

During SSW21, irreversible PV mixing driven by an exceptionally strong PW1 breaking led to the formation of a secondary
cyclone and BT–BC instability, suggesting the destabilization of WPW2 in the mid-stratosphere (YCK23). This aligns with
McIntyre's (1982) proposal that large amplitude wave breaking disrupts the basic equator-to-pole PV gradient, creating an
atmosphere characterized by scattered pieces of high and low PV. Under such conditions, energy cascades from low to high
wavenumbers across the PV and height fields to preserve enstrophy and energy. The abnormal PW1 breaking (Fig. 6b) and
the temporally out-of-phase relationship, wherein PW1 weakened as PW2 strengthened (Fig. 2b), suggest that a similar
phenomenon may have occurred during SSW02.





**Figure 7**: Time series of EPV at the (a) 1500 K, (b) 800 K, and (c) 400 K isentropic surfaces from 22 to 25 September 2002.





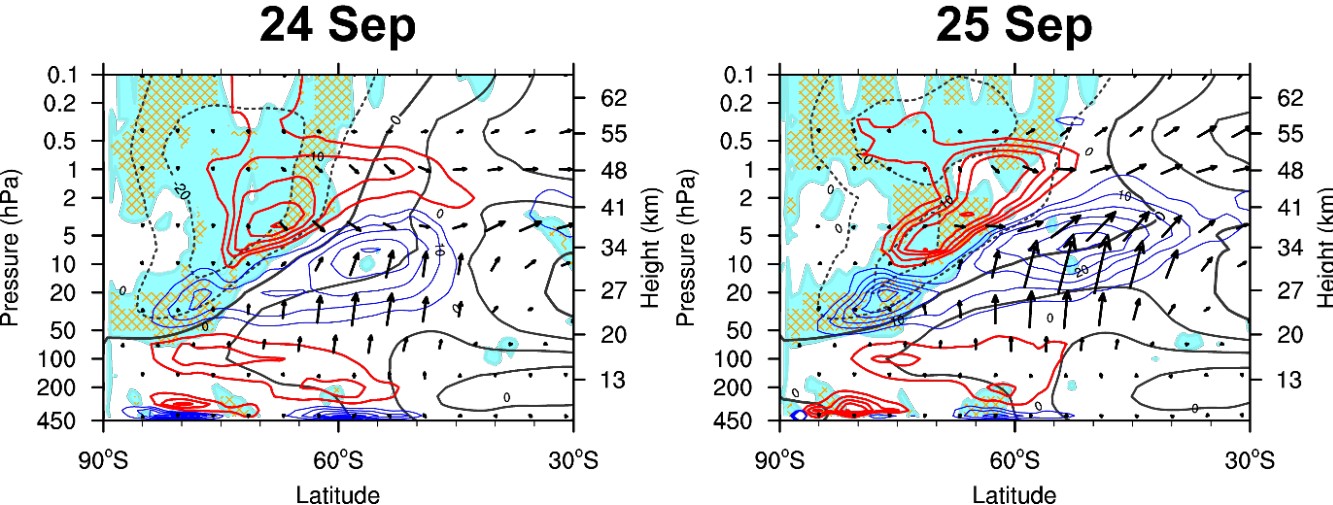

**Figure 8:** Latitude–height cross sections of the negative meridional gradient of zonally averaged quasi-geostrophic potential vorticity ($\bar{q}_y$, mint shading) and negative squared refractive index ($n^2$, orange hatching) within the instability area overlaid with PW2 EP-flux (vectors) and EPFD (contours, where red and blue indicate positive and negative values, respectively) on 24–25 September 2002. Black contours represent zonal-mean zonal wind, with contour specifications matching those in Fig. 3.



The evolution of Ertel's PV (EPV) on the 1500 K isentropic surface (approximately 2 hPa) in Fig. 7a closely resembles that
observed during SSW21 (Fig. 8 of YCK23). On 22 September, significant PW1 breaking and the resulting irreversible
mixing strongly deformed the vortex, ultimately forming an additional cyclone in its trailing region (90–120°E).
Simultaneously, low magnitude PV extended deeply into higher latitudes, crossing the pole between two high magnitude PV
cores. This indicates a localized reversal of the meridional PV gradient, destabilizing the flow. These features suggest that
PW1 breaking triggered smaller-scale wave generation through energy cascading and initiated BT–BC instability, which
could further amplify these smaller-scale waves. This process likely contributed to PW2 enhancement in the mid-to-upper
stratosphere.

Unlike SSW21, SSW02 involved the separation of the primary cyclone, which began on 23 September. Wavenumber
decomposition revealed that this separation accounted for the substantial amplification of PW2 from 24 September (not
shown). The formation of the secondary cyclone and its subsequent eastward migration at the 1500 K isentropic surface can
be traced back through the 800 K isentropic surface (approximately 10 hPa, Fig. 7b) to structures originating in the lower
stratosphere near 400 K (approximately 100 hPa, Fig. 7c). Based on earlier evidence supporting in situ WPW2 generation in
the mid-to-upper stratosphere via BT–BC instability, this upward-propagating signal suggests that unstable WPW2
excitation is associated with the incident PW2 from below. This seemingly counterintuitive interpretation can be understood
regarding the over-reflection of waves from below. The concept of over-reflection relates incident PWs to the in situ PW
excitation through BT–BC instability (Rhodes et al., 2021). As previously mentioned, a critical layer embedded within an
unstable region (with $\bar{q}_y < 0$) can be a source for unstable PW growth (Dickinson, 1973). If incident PWs can tunnel from
the turning level (where waves first become evanescent) to the critical level (where $\bar{u} = C_x$) through the evanescent region
(where $n^2 < 0$ owing to $\bar{q}_y < 0$ and $\bar{u} - C_x > 0$), these waves can grow by extracting energy from the mean flow. From this
perspective, the growth of unstable PWs is initiated by PWs tunneling beyond the turning level and amplifying at the critical
level (e.g., Harnik and Heifetz, 2007). Over-reflection occurs when an incident PW is reflected from the turning level,
gaining more energy than it originally had (Rhodes et al., 2021) and the divergence of EP-flux represents this energy growth.
Figure 1 and the related discussion in the paper by Rhodes et al. (2023) provide further details.

The possibility of over-reflection is explored in Fig. 8, which presents latitude–height sections of PW2 EP-fluxes and EPFD
along with the evanescent region (negative $n^2$, orange hatching) within the destabilized area (negative $\bar{q}_y$, cyan shading).
This figure focuses on 24–25 September, when the primary cyclone became fully detached. Here, $n^2$ is calculated by setting
ZWN $k = 2$ with a zonal phase speed $C_x$ of –20 m s$^{-1}$. As the negative $C_x$ increases, the evanescent region expands toward
the easterly core due to the increasing area where $\bar{u} - C_x > 0$, leading to $n^2 < 0$. Thus, the negative $n^2$ with the selected $C_x$
(–20 m s$^{-1}$) roughly encompasses the evanescent region derived from the major $C_x$ range of westward components of PW2
propagating upward from 100 hPa (–20–0 m s$^{-1}$, Fig.4), which are subject to over-reflection in the destabilized polar



stratosphere dominated by easterlies. Some of the upward-propagating PW2 encountered the lower boundary of the WPW2
evanescent region, where the westward components of the incident PW2 were able to tunnel through. From the critical levels
above the evanescent region, downward and equatorward PW2 fluxes emerged, increasing in magnitude with distance,
thereby creating a positive EPFD. All these features suggest that the growth of stratospheric WPW2 was associated with the
incident PW2 tunneling to their critical levels through the evanescent region and subsequent amplification at those levels. As
such, the downward propagation of PW2, opposite to the upward-propagating incident PW2, is interpreted as over-reflection,
with the positive EPFD indicating that these waves had greater energy than the incident ones. While the positive EPFD
region extends farther equatorward, the area of potential PW generation via instability remained largely confined to higher
latitudes in Fig. 8. This is likely because $\bar{q}_y$, calculated using zonal-mean variables, does not fully capture the longitudinally
localized instability extending into lower latitudes (Fig. 7a). Additionally, the excitation of unstable PWs through nonlinear
wave–wave interactions occurred simultaneously, contributing to positive EPFD. Although both wave generation
mechanisms via instability operated simultaneously, over-reflection appears to play an increasingly dominant role in
amplifying PW2 as the onset approached.

**4 Summary and Conclusion**
Since the initiation of routine upper-atmosphere observations, only one SSW has been recorded in the SH, occurring on 25
September 2002. This SSW event was marked by the splitting of the polar vortex, a phenomenon rarely observed even in the
NH. Early studies in the 2000s primarily examined the role of tropospheric PWs and vortex preconditioning, which direct
these waves toward the polar stratosphere, in triggering SSW02. However, the influence of spontaneously generated waves
within the stratosphere remains unexplored. Building on the recent findings of YCK23, which highlighted the critical role of
instability-induced stratospheric waves in vortex splitting during the 2021 NH SSW, this study revisits SSW02, focusing on
the potential contribution of in situ-excited PWs to the vortex split.

Consistent with previous studies, the substantial amplification of PW2 at 10 hPa, which led to the sudden split of the polar
vortex, can be traced back to anomalous bursts of ZWN2 waves in the troposphere. However, this study also identifies the
simultaneous descent of WPW2 from the mid-to-upper stratosphere to 10 hPa, suggesting their contribution to the vortex
split. These WPW2s were generated in situ within the polar stratosphere, which was driven toward BT–BC instability as the
zonal wind reversal progressed downward from the lower mesosphere including the WPW2 critical layer. Instability
amplified PW2 through two distinct mechanisms: nonlinear wave–wave interactions triggered by PW1 breaking – similar to
the process observed during SSW21 (YCK23) – and the over-reflection of upward-propagating PW2. As the onset
approached, over-reflection played an increasingly prominent role in PW2 enhancement. These in situ-excited WPW2
further contributed to PW2 intensification at 10 hPa through downward propagation. A double-jet configuration, previously





proposed as a vortex preconditioning mechanism for inducing instability during SSW21 (YCK23), also preceded SSW02.
However, unlike in SSW21, the critical-level interaction between the double jet and tropospheric PW1 was absent in SSW02.
Instead, an anomalous poleward shift of the polar vortex led to zonal wind reversal and vortex destabilization by confining
PW1 to the polar stratosphere and enhancing westward momentum deposition in that region.

Common insights emerge from this study and YCK23, which examined SSW events across different hemispheres:
Anomalous PW1 breaking leads to zonal wind transitions to easterlies, destabilizing the stratosphere during major SSW
development. The subsequent growth of unstable PW2 contributes to polar vortex splitting. Among the 11 wave-2-type
major SSW events exhibiting vortex split characteristics in the NH over the 44-year period from 1979 to 2023 (classified by
Ryoo and Chun 2005; Table S1), six cases present the simultaneous occurrence of PW1 dissipation, BT–BC instability, and
PW2 generation within the stratosphere (Fig. S1). Although this assessment is based on a preliminary visual inspection
disregarding time lag among these phenomena, it suggests that in situ PW2 generation via instability may have played a
more dominant role in approximately half of vortex-splitting SSW events than tropospheric wave forcing. This highlights the
critical role of explosive unstable PW growth within the stratosphere in vortex splitting, though this mechanism is not
exclusive to split cases. Given the significant impact of in situ PW2 excitation via instability on vortex morphology—a key
factor in shaping SSW characteristics and its downward influence—incorporating this mechanism into SSW research could
provide a more comprehensive understanding of SSW dynamics.

Anomalous easterlies in the equatorial upper stratosphere are another shared feature between SSW02 and SSW21. The
occurrence of both events during the westerly phase of the QBO in the lower stratosphere (50 hPa) – a condition that
typically suppresses SSWs according to the Holton–Tan effect – supports the role of equatorial upper stratospheric winds in
triggering SSWs. Notably, similar vortex shifts linked to equatorial upper stratospheric easterlies were also observed during
the 2019 SH minor warming event, although the easterlies in this case were not anomalous. Additionally, Koushik et al.
(2022) reported that equatorial easterlies in the upper stratosphere were present in approximately 70% of 29 NH SSW events
from 1979 to 2021. They further highlighted a growing frequency of SSWs preceded by this wind structure since 2000,
suggesting a shift in the system's dynamics. In this context, further research is warranted in two key areas: 1) the processes
governing the development of anomalous equatorial upper stratospheric easterlies that trigger SSWs, particularly their
connection to equatorial waves and lower stratospheric mean flows; 2) the reason underlying the increasing frequency of
SSWs with these easterlies, linked to climate change.



## Data availability

MERRA-2 data are available from the Global Modeling and Assimilation Office at NASA Goddard Space Flight Center via the NASA GES DISC online archive (https://doi.org/10.5067/WWQSXQ8IVFW8, GMAO, 2015). The data supporting this study's findings are available upon request from the corresponding authors.

## Author contributions

JHY and HYC conceived the study. JHY conducted the formal analysis and visualized the results. JHY drafted the manuscript with contributions from HYC.

## Competing interests

The authors declare no conflicts of interest.

## Financial support

This research was supported by a National Research Foundation of Korea grant funded by the South Korean government (2021R1A2C100710212).

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
