# Peer review of "Role of in situ-excited planetary waves in polar vortex splitting during the 2002 Southern Hemisphere sudden stratospheric warming event"

_EGUsphere, 2025_

## Referee Comment (RC2)

Review of egusphere-2025-748

**Summary and General Comments**

This is a well-written case study of the 2002 stratospheric sudden warming event in the southern hemisphere, referred to in the paper as SSW02. Given its extremely unusual nature, SSW02 has attracted attention in the literature for many years. The core contribution of the present study is to examine the role of in situ-excited planetary wave 2 in causing SSW02, as distinct from zonal wavenumber 2 waves propagating up from the troposphere.

I think this study merits prompt publication, albeit after what are likely to be major revisions. The single largest concern I have about the current version of the work is that the conclusions lean on the upper stratospheric and lower mesospheric fields of a single reanalysis product (MERRA-2). But how reliable are those, really? A recent study [1] of an NH SSW event found that MERRA-2 had problems above ~60 km and the only satellite instrument used by MERRA-2 above that altitude (Aura MLS) didn't launch until 2004. It's one thing to argue that the GCM underlying a reanalysis should be able to spread information from lower altitude upward, but the arguments made in the present paper involve apparent downward propagation of both waves and the zonal jet from higher altitudes. This would seem to create a greater need for high-altitude data.

Unfortunately I am not clear on how good the observing system for this part of the atmosphere was in September 2002. A recent whole-atmosphere reanalysis [2] starts in 2004 and used 6 satellite instruments, only one of which (TIMED SABER, not used in MERRA-2) was active in 2002. A technical report about the MERRA-2 observing system [3] warns of large biases in the lower mesosphere of the model, although it also suggests that the reanalysis state at those altitudes should be influenced by (if nothing else) channel 14 of the AMSU-A which was carried on three NOAA satellites active in 2002. In contrast, the ERA5 description paper [4] refers to the mesosphere as "observation-free" and warns of associated artifacts in the ERA5 product. However, ERA5 is also assimilating AMSU-A channel 14.

Within the framework of the present manuscript, I think the best way to address this issue is to repeat at least some of the major analyses using a different reanalysis product to see how robust the results are. It would be fine to skip the statistical significance calculations (about which more below), in which case the amount of new data that needs to be downloaded will not be large. It's not obvious to me which reanalysis should be used for the comparison—ERA5 is actually available in two versions (5 and 5.1) for the early 2000s for reasons laid out in [4] and an associated ECMWF technical report [5]. (Note that the technical report includes some figures showing SSW02 in several different ECMWF reanalyses.) Perhaps one of the Japanese reanalyses would be appropriate instead. If the two reanalyses agree, great! If not, I think the study is still worth publishing as a demonstration of the limits of existing reanalysis for studies of this event and as a prompt for future research.

**Specific Comments By Line Number**

**24-25** One of your references (Newman and Nash 2005) attracted a (2014!) comment claiming that there was an SH SSW in 1972 and citing some related works—do those works need to be cited and/or your claim modified?

**26** "breakdown of the polar vortex during midwinter"→the event took place on 25 September, but to me at least SH winter means JJA. So I would classify the event as happening in spring.

**27** I don't quite understand what is meant by "sparse mountainous".

**78** The MERRA-2 model top is actually at 0.01 hPa (not 0.1 hPa), although I gather you are using the pressure level data that ends at 0.1 hPa.

**78-79** Sentence reads as though MERRA-2 ends at the end of 2023, which is not correct. What you mean is that you're using the 44 years 1980-2023 to calculate the characteristic magnitude of variability used to contextualize the size of SSW02.

**84** Make sure to double-check this equation to make sure it has correct factors of cos(latitude)

**94** This equation and subsequent discussions of BT-BC instability involve derivatives with respect to "y"—this would make sense for theoretical analyses on the QG beta plane, but you're analyzing real data on the sphere. You should either explain what you mean by "y" or just rewrite this equation, references to $q_y$, etc. in the physically correct spherical geometry.

**96-97** This is an allusion to the way in which the flow is satisfying the Charney-Stern-Pedlosky criterion for instability, right? Maybe you should say that directly. It might also be interesting to point out that this isn't necessarily the most common way to satisfy the CSP criterion—see discussion in [6].

**101** The cos(latitude) superscript appears to be a typo.

**101-107** These equations use both f and $f_0$—the generalization of QG theory to the sphere is not trivial, and (assuming these equations are all written correctly) you should clearly state what you have chosen for $f_0$ and (in my opinion) say a few words to remind readers about why you are using both f and $f_0$.

**115** Just for clarity maybe you should state that k is the nondimensional zonal wavenumber.

**122** "the PNJ weakened dramatically by more than 100 m/s"—is the weakening really *that* large? I'm not seeing it in the figure.

**124** "statistically significant at the 99% confidence level"—in my opinion "statistical significance" is an odd concept to apply to a case study of a single event that is well-known to be both a) real and b) unusual. In this context there is no "null hypothesis" being tested, you're just trying to quantify how extreme SSW02 was. I think you should drop the language here and elsewhere in the paper about "statistical significance" and "confidence intervals" and just state that the various colored dots represent <0.5th/<2.5th/>97.5th/>99.5th percentile as appropriate.

**126** Are we supposed to be able to see the claimed upward-propagating signal in the figure? I also don't understand how this claim is supposed to related to the downward propagation of the easterly wind from the mesosphere (or even if the claims are supposed to be related).

**Figure 2a** This figure might be easier to understand if you also had a supplementary figure showing the full geopotential height field at 10 hPa on these same dates, instead of just the GHP. Figure 7 does sort of do this, but for somewhat different levels and times.

**Figure 3** This figure seems to be (in the preprint at least) a raster graphic. I think you should try to have this figure (and all others) published as vector graphics, to enable zooming in to view the many details.

**158-159** This sentence reads to me as claiming that there were easterlies in 70S-50S, <10 hpa on 22 September, which does not actually appear to be true.

**Figure 3a** How is the orientation of the EP flux vectors selected in this and all other EP flux figures. This isn't a trivial issue [7].

**170-173** Seems plausible, but did you actually check plots of separate $q_y$ subterms to confirm?

**173** The overlap isn't exact, though—is it worth commenting on why that might be?

**193-194** The caption should clarify that it's the *zonal wavenumber 2* component of Z' that is being plotted in Figure 3c.

**Figure 4** The main text about and/or caption of this figure need much more discussion of how the phase speed spectra are computed—e.g., doesn't a phase speed spectrum need to be computed over some finite-width time window? What was that window and how was it selected, particularly given that you are attempting to interpret short-term temporal variations of the diagnosed phase speeds. If the analysis follows some previous paper sufficiently closely, you could just cite that paper instead.

**215** What "decreasing trend at 100 hPa"? (I don't see it in the figure.)

**222** How sure are you about the 21 September date, given the finite width of the contour interval? (In other words, might the jet structure you describe have been visible on 20 September if the upper left panel of Figure 5 were redrawn with a lot more contours?)

**226** Technically there was a very small region of easterlies on 21 September.

**242** I don't really understand what is meant by "negative PW forcings". Also maybe you should include the figure(s) you refer to as "not shown" in the supplementary material?

**246-247** I guess near the jet core the EPFD is at or below the 2.5th percentile value, but aren't more of the values actually below the 0.5th percentile?

**278-279** If the "additional cyclone" is what I think it is, might not its location be better described as 90E-180E?

**288-292** Are you sure about these claims? (I have a hard time seeing them in the figure.)

**Figure 8** The figure panels contain a bunch of zeros that don't seem to be within (plotted) contour lines. What are they for? Maybe you should plot the associated zero contours?

**306** Clarify that "increases" means "increases towards zero", assuming that is what is meant.

**310-311** Are we supposed to be able to see EP fluxes going upwards at the bottom of the $n^2<0$ region?

**318-319** You're arguing that the possibility of instability is sensitive to the zonally resolved—not just zonal mean—$q_y$, right? That doesn't necessarily seem impossible, but if you're thinking about the instability of *zonal wavenumber 2* doesn't this wave have to "feel" the PV gradient over a pretty zonally extended region (perhaps even all longitudes) precisely because its wavelength is so long?

**320-322** Sorry, I don't understand what is substantiating this claim. (Maybe I just don't understand over-reflection well enough?)

**Figure S1/Table S1** Thank you for including this information, but I think it's sort of weird to refer to 6 pages of plots as a single "figure". I think they should just be renumbered as 6 different figures.

**352-354** Can you explain a bit more about how you are concluding that "in situ PW2 generation via instability may have played a more dominant role in approximately half of vortex-splitting SSW events than tropospheric wave forcing"?

**356-358, 367-369** Maybe you should add a line to your abstract saying that you discussion the implications of your (SSW02-specific) results for our understanding of SSW dynamics in general.

**364** What exactly do you mean by "not anomalous"? Surely the easterlies weren't exactly at their climatological mean values?

**369-370** I imagine this will be tough to do with reanalysis, given the sparse and time-dependent nature of the observing system at some of the relevant altitudes. If this really is a forced response to climate change, hopefully it can be modeled robustly.

**Additional Thoughts on Future Work**
Regarding the long-term possibility of (much more challenging) additional work, unfortunately there doesn't seem to be much satellite data not already included in MERRA-2 or ERA5. However MIPAS on Envisat and SMR on Odin may have usable data—indeed, a MIPAS-based study of SSW02 was published nearly two decades ago [8]. (As explained in [2], the aforementioned SABER switches between observing 53S-83N and 83S-53N every 60 days and a quick look at the SABER data shows that a 53S-83N observing period began on 19 September 2002.) I am not sure if anyone has tried assimilating MIPAS or SMR data for this event.

**References**

**[1]** Liu et al. 2025, Dynamical Response of the Middle and Upper Atmosphere to the February 2018 Sudden Stratospheric Warming Revealed by MERRA-2 and SABER, JGR-Space Physics, https://doi.org/10.1029/2024JA033528

**[2]** Koshin et al. 2025, The JAGUAR-DAS whole neutral atmosphere reanalysis: JAWARA, Prog. Earth Pla. Sci., https://doi.org/10.1186/s40645-024-00674-3

**[3]** McCarty et al. 2016, MERRA-2 Input Observations: Summary and Assessment, GMAO Tech. Report, https://ntrs.nasa.gov/citations/20160014544

**[4]** Hersbach et al. 2020, The ERA5 global reanalysis, QJRMS, https://doi.org/10.1002/qj.3803

**[5]** Simmons et al. 2020, Global stratospheric temperature bias and other stratospheric aspects of ERA5 and ERA5.1, ECMWF Tech. Report, https://doi.org/10.21957/rcxqfmg0

**[6]** Vallis 2006, Atmospheric and Oceanic Fluid Dynamics: Fundamentals and Large-scale Circulation, Cambridge University Press, p. 265/equation 6.74, available via Google Books

**[7]** Jucker 2021, Scaling of Eliassen-Palm flux vectors, Atmos. Sci. Lettrs., https://doi.org/10.1002/asl.1020

**[8]** Wang et al. 2005, Longitudinal variations of temperature and ozone profiles observed by MIPAS during the Antarctic stratosphere sudden warming of 2002, JGR-Atmospheres, https://doi.org/10.1029/2004JD005749

---

## Author Response (AR1)

**Response to Reviewers' Comments**

**Ji-Hee Yoo and Hye-Yeong Chun**

Department of Atmospheric Sciences, Yonsei University, Seoul, Korea

17 July, 2025

Dear Editor and Reviewers,

Thank you very much for your constructive comments and suggestions on this paper. We carefully revised the manuscript to address all comments and made every effort to improve its clarity and scientific rigor. In particular:

- I. We discussed the potential dependency between two instability-driven in-situ PW2 growth mechanisms—nonlinear wave—wave interactions and over-reflection—and their differing temporal contributions to PW2 amplification.
- II. We supplemented our analysis with two additional reanalysis datasets, to support the robustness of our results involving the mesosphere, where reanalysis reliability is limited due to sparse observation.
- III. We incorporated a discussion of the findings from O'Neill et al. (2017), whose analysis of SSW02 underscores the importance of nonlinear PV advection and vortex–vortex interactions in vortex splitting, beyond the classical wave—mean flow framework adopted in this study.

We respond to each reviewer's comments in the following paragraphs. The original comments are shown in blue, and our responses are given in black.

We hope that the revised manuscript meets the journal's standards and will be considered for publication in *Atmospheric Chemistry and Physics*.

Sincerely,

Hye-Yeong Chun

**Reviewer #1's Comments**

**General comment:**

In this paper, the authors examine the role of in situ exited planetary waves (PWs) for the major sudden stratospheric warming in the Southern Hemisphere 2002. It demonstrates that westward propagating PWs with zonal wave number 2 (PW2) were amplified due to the barotropic-baroclinic instability in the stratosphere and split the polar vortex. The instability was formed by the breaking of zonal wave number 1 PWs and depositing westward momentum. Furthermore, the authors suggest that the over-reflection of upward propagating PW2 contributed to split the polar vortex.

The paper is well-written and is highly relevant to EGUsphere. While some minor revisions are needed, I recommend the paper for publication, pending a few minor revisions.

Thank you very much for carefully reviewing the manuscript and providing valuable comments on the manuscript. The comments and suggestions are very helpful, and we made best effort to address them during the revision process to improve the manuscript.

**Minor Comments:**

1. Sec 3.5, L288: ... can be traced back ... It is hard to identify from Figure 7. Please specify more clearly.

Thank you for the helpful comment. As you pointed out, the upward development process described in the original sentence was not sufficiently clear based on Fig. 7 alone. To address this, we examine the geopotential height perturbations (GHP) at 100, 10, and 2 hPa (Fig. R1.1), which approximately correspond to the isentropic surfaces of Ertel's potential vorticity (EPV) shown in Fig. 7.

As shown in Fig. R1.1 below, at 100 hPa, a clear zonal wavenumber (ZWN) 2 pattern appears in GHP from 22 September. At 10 hPa, the primary cyclone becomes pinched into a peanut-like structure on 23 September, and by 24 September, it splits into two cyclones. At 2 hPa, these processes occur with a one-day delay, as the pinch forms on 24 September and the separation becomes distinct from 25 September onward, suggesting that the vortex split initiates in the lower stratosphere and subsequently develops upward. This progression is consistent with the vertical evolution of PW2 amplitude (Fig. 2b), where lower stratospheric enhancement precedes that of the upper stratosphere, and with the westward tilt of the phase of PW2 with height up to 3 hPa (Fig. 2c).

We included this description in the revised manuscript (L303–306), with Fig. R1.1 added to the supplementary material (Fig. S3) for further clarification.

Figure R1.1. Polar stereographic series of GHP at (a) 2 hPa, (b) 10 hPa, and (c) 100 hPa from 22 to 25 September 2002.

2. Sec 3.5, from L286: Two paragraphs describe the possibility of over-reflection, which amplified the incident PW2 from troposphere. But main component of incident PW2 was EPW2, although amplified PW2 was mainly WPW2, as shown in Fig.4. It is OK? Please clarify.

Thank you for bringing this point to our attention. We originally thought that although incident PW2 was predominantly eastward-propagating ones (EPW2) at 100 hPa, the small portion of westward-propagating PW2 (WPW2) were amplified by instability as their critical levels reside in the instability region. However, during the revision process, we found that EPW2 hardly reach the upper stratosphere, as they encounter background winds increasingly unfavorable for upward propagation—where westerlies transition to easterlies with height (Fig. 3), according to the Charney and Drazin (1961) criterion  $[0

**Figure R2.1.** Time-height cross sections of the zonal-mean zonal wind at 60°S (left) and polar cap temperature averaged over 60–90°S (right) revealed in (a) MERRA-2, (b) JRA-3Q, and (c) ERA5.1.

**Figure R2.2.** Time-height cross sections of GHP amplitude of PWs with ZWN1 (PW1, left) and 2 (PW2, right) at 60°S revealed in (a) MERRA-2, (b) JRA-3Q, and (c) ERA5.1.

**Figure R2.3.** Latitude—height cross sections of the EP-flux (vectors) overlaid on EPFD (shading) for PW2 from 22 to 25 September 2002 revealed in (a) MERRA-2, (b) JRA-3Q, and (c) ERA5.1.

**Figure R2.4.** Negative  $\bar{q}_y$  (colors) overlaid with the positive EPFD of PW2 (red contours) from 22 to 25 September 2002 revealed in (a) MERRA-2, (b) JRA-3Q, and (c) ERA5.1.

**Figure R2.5.** Latitude—height cross sections of EP-flux (vectors) overlaid on EPFD (shading) for PW1 from 20 to 25 September 2002 revealed in (a) MERRA-2, (b) JRA-3Q, and (c) ERA5.1.

**Specific Comments By Line Number:**

1. 24-25 One of your references (Newman and Nash 2005) attracted a (2014!) comment claiming that there was an SH SSW in 1972 and citing some related works—do those works need to be cited and/or your claim modified?

We have reviewed the (2014!) comment on Newman and Nash (2005). The studies cited there by Sehra (1975, 1976, 1979) reported a sharp stratospheric warming based on rocket soundings at Molodezhnaya Station (67°40'S, 45°51'E). However, in our manuscript, "major sudden stratospheric warming" (SSW) refers to events that satisfy the WMO/IQSY (1964) criterion, which requires both a significant polar stratospheric temperature increase and a reversal of the zonal-mean zonal wind from westerlies to easterlies at 60°N and 10 hPa, though the application of this definition to the Southern Hemisphere remains debated.

Since temperature observations from a single station are insufficient to confirm the occurrence of a major SSW, we examined this possibility for August–September 1972, when Sehra reported the warming, using JRA-3Q data. The zonal-mean zonal wind at 60°S and 10 hPa (Fig. R2.6) showed no wind reversal, indicating no major SSW occurred. Thus, we retain the original description in our manuscript.

Figure R2.6. Time series of zonal-mean zonal wind at 60°S and 10 hPa during August-September 1972.

2. 26 "breakdown of the polar vortex during midwinter"→the event took place on 25 September, but to me at least SH winter means JJA. So I would classify the event as happening in spring.

We modified "midwinter" to "spring" as suggested (L30).

3. 27 I don't quite understand what is meant by "sparse mountainous".

The original expression was intended to convey the reduced mountainous land cover. We revised that to "less mountainous terrain" for clarity (L31).

4. 78 The MERRA-2 model top is actually at 0.01 hPa (not 0.1 hPa), although I gather you are using the pressure level data that ends at 0.1 hPa.

You are right. We modified the original sentence in the revised manuscript (L81) for clarity.

5. 78-79 Sentence reads as though MERRA-2 ends at the end of 2023, which is not correct. What you mean is that you're using the 44 years 1980-2023 to calculate the characteristic magnitude of variability used to contextualize the size of SSW02.

Thank you for the comment. We revised the original sentence to avoid potential misunderstanding (L83).

6. 84 Make sure to double-check this equation to make sure it has correct factors of cos(latitude)

Thank you for the comment. We carefully double-checked the EP-flux formulation, and confirmed that all factors of cos (latitude) are consistent with the standard spherical-coordinate expression provided in Andrews et al. (1987, Eq. 3.5.3).

7. 94 This equation and subsequent discussions of BT-BC instability involve derivatives with respect to "y"—this would make sense for theoretical analyses on the QG beta plane, but you're analyzing real data on the sphere. You should either explain what you mean by "y" or just rewrite this equation, references to qy, etc. in the physically correct spherical geometry.

Thank you for pointing this out. We rewrote Eq. (3) in the spherical geometry (L98). However, we retain the compact notation  $\bar{q}_y$  throughout the manuscript, with the clarification that  $y = a\phi$  (L102).

8. 96-97 This is an allusion to the way in which the flow is satisfying the Charney-Stern-Pedlosky criterion for instability, right? Maybe you should say that directly. It might also be interesting to point out that this isn't necessarily the most common way to satisfy the CSP criterion—see discussion in [6].

You are correct. The reversal of  $\bar{q}_y$  sign corresponds to one of the Charney-Stern-Pedlosky (CSP) necessary conditions for instability, which requires that at least one of the following criteria be satisfied: (i)  $Q_y$  changes sign within the domain, (ii)  $Q_y$  has the opposite sign to  $U_z$  at the upper boundary, (iii)  $Q_y$  and  $U_z$  at the lower boundary have the same sign, or (iv)  $U_z$  has the same sign at both boundaries when  $Q_y = 0$  (Vallis, 2017). Here, U and Q are the zonal-mean zonal wind and potential vorticity, respectively. We state this in the revised manuscript without detailing all four criteria.

In addition, we include a brief note that, under typical midlatitude conditions, instability criterion is normally satisfied through (iii), where both  $Q_y$  and  $U_z$  at the lower boundary are positive (L103–105).

9. 101 The cos(latitude) superscript appears to be a typo.

It is corrected in the revised manuscript (L108).

10. 101-107 These equations use both f and f0—the generalization of QG theory to the sphere is not trivial, and (assuming these equations are all written correctly) you should clearly state what you have chosen for f0 and (in my opinion) say a few words to remind readers about why you are using both f and f0.

Thank you for your careful observation. You are correct — the appearance of  $f_0$  in the equations was a typo. We corrected the equations to consistently use f in these equations (L98, L108–109).

11. 115 Just for clarity maybe you should state that k is the nondimensional zonal wavenumber.

Thank you for pointing this out. We clarified that k refers to the nondimensional zonal wavenumber (L120).

12. 122 "the PNJ weakened dramatically by more than 100 m/s"—is the weakening really *that* large? I'm not seeing it in the figure.

Thank you for bringing this to our attention. Upon re-examination, we found that the actual weakening was approximately 57 m/s. We corrected this in the revised manuscript (L128).

13. 124 "statistically significant at the 99% confidence level"—in my opinion "statistical significance" is an odd concept to apply to a case study of a single event that is well-known to be both a) real and b) unusual. In this context there is no "null hypothesis" being tested, you're just trying to quantify how extreme SSW02 was. I think you should drop the language here and elsewhere in the paper about "statistical significance" and "confidence intervals" and just state that the various colored dots represent <0.5th/<2.5th/>97.5th/>99.5th percentile as appropriate.

Thank you for the helpful suggestion. As you recommended, we eliminated phrases such as statistical significance and confidence level throughout the manuscript, replacing them with percentile-based description to quantify the extremity of SSW02 in an objective manner (L130–131, L142, L154–155, L165–166, L168, L255–256, L258–259).

14. 126 Are we supposed to be able to see the claimed upward-propagating signal in the figure? I also don't understand how this claim is supposed to related to the downward propagation of the easterly wind from the mesosphere (or even if the claims are supposed to be related).

Thank you for the comment. We interpreted the upward extension of anomalously strong zonal wind deceleration (pink dots) and temperature increase (green dots) as an "upward-propagating signal," based on their evolution in the time—height cross-section (Fig. 1). However, we agree that the link between this signal and the downward propagation of easterly winds from the mesosphere is unclear. To avoid potential confusion, we removed the sentence in the revised manuscript.

15. Figure 2a This figure might be easier to understand if you also had a supplementary figure showing the full geopotential height field at 10 hPa on these same dates, instead of just the GHP. Figure 7 does sort of do this, but for somewhat different levels and times.

Figure R2.7 presents the full geopotential height field at 10 hPa for the same dates as shown in Fig. 2a. Since the spatial patterns closely resemble those in Fig. 2a, we decided not to include this figure in the supplementary material.

Figure R2.7. Polar stereographic series showing the geopotential height field at 10 hPa on 21, 23, and 25 September 2002.

16. Figure 3 This figure seems to be (in the preprint at least) a raster graphic. I think you should try to have this figure (and all others) published as vector graphics, to enable zooming in to view the many details.

Thank you for the suggestion. We converted all figures to vector graphic format in the revised manuscript.

17. 158-159 This sentence reads to me as claiming that there were easterlies in 70S-50S,

**Figure R2.8.** Latitude–height cross sections of (a–c)  $\bar{q}_y$ , the sum of first two terms (the meridional gradient of f and barotropic term), and the third term (baroclinic term) on the righthand side of Eq. (R1) and (d–f) the three terms on the right-hand side of Eq. (R2) divided by  $f^2$ , (g–j) the inverse squared Brunt–Väisälä frequency  $1/N^2$ , the vertical shear of zonal wind  $\bar{u}_z$ , the vertical gradient of Brunt–Väisälä frequency  $dN^2/dz$ , and the vertical curvature of zonal wind  $\bar{u}_{zz}$ , on 24 September 2002. The black contours represent the zonal-mean zonal winds. The solid, dashed, and thick solid lines denote positive, negative, and zero wind, respectively.

Figure R2.8a–c present the latitude–height cross sections of  $\bar{q}_y$ , the sum of first two terms (the meridional gradient of f and barotropic term), and the third term (baroclinic term) on the righthand side of Eq. (R1) on 24 September—a representative case during the vortex destabilization period (22–25 September).

The strengthening easterlies induce a positive curvature of zonal wind (barotropic term) along their core, which dominates over the meridional gradient of f, leading to a negative sum of the two terms (Fig. R2.8b). Simultaneously, the baroclinic term becomes negative both below (50–5 hPa) and above (2–0.5 hPa) the easterly jet core in the polar stratosphere (60°–90°S, Fig. R2.8c).

To clarify the mechanisms driving the negative baroclinic term, we further expand it as follows (Yoo et al. 2023):

$$-\frac{1}{\rho_0} \left( \rho_0 \frac{f^2}{N^2} \bar{u}_z \right)_z = f^2 \left[ \frac{1}{H} \frac{1}{N^2} \bar{u}_z + \frac{1}{N^4} \frac{dN^2}{dz} \bar{u}_z - \frac{1}{N^2} \bar{u}_{zz} \right]. \tag{R2}$$

Figure R2.8d–f display the first, second, and third subterms of Eq. (R2), respectively, divided by  $f^2$ . Both the first (Fig. R2.8d) and third terms (Fig. R2.8f) contribute to the negative baroclinic term within the developing easterlies over the polar stratosphere (Fig. R2.8c). Figure R2.8g–j present the distributions of four variables involved in the three subterms of Eq. (R2): the inverse squared Brunt–Väisälä frequency  $(1/N^2)$ , the vertical shear of zonal wind  $(\bar{u}_z)$ , the vertical gradient of Brunt–Väisälä frequency  $(dN^2/dz)$ , and the vertical curvature of zonal wind  $(\bar{u}_{zz})$ , respectively. Under positive  $1/N^2$  (Fig. R2.8g), the first subterm (Fig. R2.8d) becomes negative due to the easterlies descending from the lower mesosphere, which generate negative vertical shear  $(\bar{u}_z)$  along and below the jet core (Fig. R2.8h). The third subterm (Fig. R2.8f), showing distinct negative values above (2–0.5 hPa) and below (50–5 hPa) the easterly core, is attributed to strong positive vertical curvature  $(\bar{u}_{zz})$  in these regions with positive  $1/N^2$  (Fig. R2.8g, j).

In summary, the negative baroclinic term arises primarily from two processes: negative  $\bar{u}_z$  beneath and near the easterly jet core, and enhanced positive  $\bar{u}_{zz}$  both above and below the jet core. We included Figure R2.8 along with its detailed analysis in the supplementary material as Fig. S1 and Text S1, and are explicitly cited in the revised manuscript (L177–178).

**20. 173 The overlap isn't exact, though—is it worth commenting on why that might be?**

You are right—we changed the expression "largely" to "partially" (L178). As a possible explanation, we suggest that zonally localized instability could serve as a source of PW2 at lower latitudes (45–60°S), where positive EPFD appears without negative  $\bar{q}_y$  (Fig. 3b in the original manuscript), as  $\bar{q}_y$ , derived from zonal-mean fields, may not capture such zonal asymmetries. Since this inference is based on the Ertel PV distribution in Fig. 7, we mentioned it after the description of Fig. 8 (L337–341).

**21. 193-194 The caption should clarify that it's the zonal wavenumber 2 component of Z' that is being plotted in Figure 3c.**

Thank you for pointing this out. We clarified in the revised manuscript that Fig. 3c shows the zonal wavenumber 2 component of Z' (L197).

22. Figure 4 The main text about and/or caption of this figure need much more discussion of how the phase speed spectra are computed—e.g., doesn't a phase speed spectrum need to be computed over some finite-width time window? What was that window and how was it selected, particularly given that you are attempting to interpret short-term temporal variations of the diagnosed phase speeds. If the analysis follows some previous paper sufficiently closely, you could just cite that paper instead.

In calculating the phase speed spectra of PW2, we followed Song et al. (2020) using an 11-day window. Prior to determining the window size, we computed the power spectral density (PSD) of the PW2 geopotential height amplitude at 1, 10, and 100 hPa over 45–75°S during September 2002 to identify the dominant periods (Fig. R2.9). The red dashed lines in this figure indicate the red-noise background based on a first-order Markov process.

Figure R2.9. Power spectral density (PSD) of the GHP amplitude of PW2 during September 2002 at 1, 10, and 100 hPa.

At 100 hPa, a dominant peak is found at 15 days, while at 10 hPa, an additional peak is observed at 6 days. At 1 hPa, peaks are confined to periods shorter than 10 days, indicating enhanced high-frequency variability with increasing altitude. Although a longer window would better capture the dominant low-frequency, especially at 100 hPa, it may introduce uncertainties at higher altitudes due to the rapid changes in the mean fields during the evolution of the SSW02 event. Moreover, applying different window lengths at each level complicates direct vertical comparisons. Therefore, to balance the different temporal variability while ensuring consistency across vertical levels, we adopt an 11-day window as in Song et al. (2020), with sine and cosine tapering applied to the first and last 3 days, respectively, to reduce edge effects. The reference is provided in the caption of Fig. 4 in the revised manuscript (L205–207).

Meanwhile, in the process of revisiting this analysis, we found a minor error in the application of the time window. We corrected the calculation accordingly and updated Fig. 4 with the properly computed phase speed spectra of PW2. We confirm that this correction does not alter the central result.

**23. 215 What "decreasing trend at 100 hPa"? (I don't see it in the figure.)**

Thank you for the comment. The "decreasing trend at 100 hPa" refers to the reduction in the westward phase speed range from approximately 40 m/s on 21 September to around 20 m/s on 25 September.

However, the dominant WPW2 at 100 hPa with amplitudes exceeding 100 m exhibit an increasing trend with time, consistent with the patterns observed at 1 and 10 hPa. Accordingly, we removed the sentence.

**24. 222 How sure are you about the 21 September date, given the finite width of the contour interval? (In other words, might the jet structure you describe have been visible on 20 September if the upper left panel of Figure 5 were redrawn with a lot more contours?)**

Figure R2.10 presents the zonal-mean zonal wind from 1 to 20 September, using a finer contour interval of 5 m/s. As the reviewer pointed out, the double-westerly jet–like configuration was already present prior to 21 September. Accordingly, we revised the corresponding sentence in the manuscript (L234–235).

**Figure R2.10.** Latitude—height cross sections of zonal-mean zonal wind in the SH on selected dates between 1 and 20 September 2002, chosen to represent key stages in the evolution of the flow.

**25. 226 Technically there was a very small region of easterlies on 21 September.**

Thank you for pointing this out. We revised the date of the initial easterly appearance to 21 September (L238).

**26. 242 I don't really understand what is meant by "negative PW forcings". Also maybe you should include the figure(s) you refer to as "not shown" in the supplementary material?**

Thank you for the helpful comment. The term negative PW forcing refers to wave forcing necessary for reversing westerlies into easterlies. To avoid confusion, we modified the expression to explicitly describe its role in reversing the westerlies. Additionally, the comparison between total PW EPFD and PW1 EPFD, which was previously mentioned as "not shown," is now included in the supplementary material (Fig. S2) and cited accordingly in the revised manuscript (L255).

**27. 246-247 I guess near the jet core the EPFD is at or below the 2.5th percentile value, but aren't more of the values actually below the 0.5th percentile?**

Thank you for pointing out this. As you correctly noted, a substantial portion of the negative EPFD values falls below the 0.5th percentile near the jet core. We revised the corresponding sentence in the revised manuscript (L256).

**28. 278-279 If the "additional cyclone" is what I think it is, might not its location be better described as 90E-180E?**

Thank you for the comment. We revised the location description to 90°E–180° accordingly (L292).

**29. 288-292 Are you sure about these claims? (I have a hard time seeing them in the figure.)**

Thank you for your comment. This issue was also raised by Reviewer 1 (Comment #1). Please refer to our response there.

**30. Figure 8 The figure panels contain a bunch of zeros that don't seem to be within (plotted) contour lines. What are they for? Maybe you should plot the associated zero contours?**

Thank you for the helpful comment. The zeros correspond to zero EPFD. To avoid any confusion, we removed all line labels associated with EPFD and retained line labels only for the zonal-mean zonal wind in this figure. In addition, we added a clarification that the contour intervals for EPFD follow the color bar in Fig. 3a (L287).

**31. 306 Clarify that "increases" means "increases towards zero", assuming that is what is meant.**

"Increases" refers to a rise of  $C_x$  toward higher amplitude values. We modified the expression accordingly to avoid confusion (L324).

**32. 310-311 Are we supposed to be able to see EP fluxes going upwards at the bottom of the n2**

Figure R2.11. Meridional gradient of EPV at the 1500 K isentropic surface in 23–24 September 2002.

**34. 320-322 Sorry, I don't understand what is substantiating this claim. (Maybe I just don't understand over-reflection well enough?)**

Thank you for your comment. This question was also raised by Reviewer 1 (Comment #3). Please refer to our response there.

35. Figure S1/Table S1 Thank you for including this information, but I think it's sort of weird to refer to 6 pages of plots as a single "figure". I think they should just be renumbered as 6 different figures.

Thank you for the suggestion. As recommended by the reviewer, we renumber Fig. S1 as six separate figures (Fig. S5–S10, L389).

36. 352-354 Can you explain a bit more about how you are concluding that "in situ PW2 generation via instability may have played a more dominant role in approximately half of vortex-splitting SSW events than tropospheric wave forcing"?

Thank you for your insightful comment. Based on this study and our previous work (Yoo et al. 2023), we evaluated the occurrence of instability-induced PW2 across wave-2 vortex splitting events by establishing three indicative criteria—(1) PW1 dissipation (negative EPFD), (2) the emergence of instability (negative  $\bar{q}_y$ ), and (3) PW2 generation (positive EPFD). However, more detailed case-by-case analyses are needed to assess the relative contribution of each process. Accordingly, we revised the relevant sentence and softened expressions elsewhere in the manuscript that previously described the role of this mechanism as being significant (L390–393).

37. 356-358, 367-369 Maybe you should add a line to your abstract saying that you discussion the implications of your (SSW02-specific) results for our understanding of SSW dynamics in general.

Thank you for the helpful suggestion. We revised the abstract to include the broader implications of our SSW02-specific findings for the general understanding of SSW dynamics (L20–22).

38. 364 What exactly do you mean by "not anomalous"? Surely the easterlies weren't exactly at their climatological mean values?

The term "not anomalous" in this context means that the easterlies did not qualify as a statistical anomaly. However, we agree that this expression is not essential in the current context and could lead to confusion. Therefore, we removed the phrase in the revised manuscript.

39. 369-370 I imagine this will be tough to do with reanalysis, given the sparse and time-dependent nature of the observing system at some of the relevant altitudes. If this really is a forced response to climate change, hopefully it can be modeled robustly.

Thank you for your constructive comment. We reflected your suggestion in the revised manuscript (L406).

**Additional Thoughts on Future Work**

Regarding the long-term possibility of (much more challenging) additional work, unfortunately there doesn't seem to be much satellite data not already included in MERRA-2 or ERA5. However MIPAS on Envisat and SMR on Odin may have usable data—indeed, a MIPAS-based study of SSW02 was published nearly two

decades ago [8]. (As explained in [2], the aforementioned SABER switches between observing 53S-83N and 83S-53N every 60 days and a quick look at the SABER data shows that a 53S-83N observing period began on 19 September 2002.) I am not sure if anyone has tried assimilating MIPAS or SMR data for this event.

Thank you for the thoughtful suggestion. We agree that further investigation of SSW02 using reanalyses incorporating satellite data such as MIPAS or SMR may offer valuable additional insights. Although this is beyond the scope of the present study, it could be a valuable direction for future research.

**References**

- [1] Liu et al. 2025, Dynamical Response of the Middle and Upper Atmosphere to the February 2018 Sudden Stratospheric Warming Revealed by MERRA-2 and SABER, JGR-Space Physics, https://doi.org/10.1029/2024JA033528
- [2] Koshin et al. 2025, The JAGUAR-DAS whole neutral atmosphere reanalysis: JAWARA, Prog. Earth Pla. Sci., https://doi.org/10.1186/s40645-024-00674-3
- [3] McCarty et al. 2016, MERRA-2 Input Observations: Summary and Assessment, GMAO Tech. Report, https://ntrs.nasa.gov/citations/20160014544
- [4] Hersbach et al. 2020, The ERA5 global reanalysis, QJRMS, https://doi.org/10.1002/qj.3803
- [5] Simmons et al. 2020, Global stratospheric temperature bias and other stratospheric aspects of ERA5 and ERA5.1, ECMWF Tech. Report, https://doi.org/10.21957/rexqfmg0
- [6] Vallis 2006, Atmospheric and Oceanic Fluid Dynamics: Fundamentals and Large-scale Circulation, Cambridge University Press, p. 265/equation 6.74, available via Google Books
- [7] Jucker 2021, Scaling of Eliassen-Palm flux vectors, Atmos. Sci. Lettrs., https://doi.org/10.1002/asl.1020
- [8] Wang et al. 2005, Longitudinal variations of temperature and ozone profiles observed by MIPAS during the Antarctic stratosphere sudden warming of 2002, JGR-Atmospheres, https://doi.org/10.1029/2004JD005749

**Reviewer #3's Comments**

**General comment:**

Review of "Role of in situ-excited planetary waves in polar vortex splitting during the 2002 Southern Hemisphere sudden stratospheric warming event", authored by Ji-Hee Yoo and Hye-Yeong Chun.

This manuscript presents a study on the dynamics of the only major sudden warming that has been recorded on the Southern Hemisphere, using the reanalysis MERRA2. The authors thoroughly analyze the 5-day period before the central date of the event in terms of (linear) wave-mean flow interaction. They conclude that the breaking of planetary-scale wavenumber 1 in the polar stratosphere destabilized the flow and contributed to the generation of smaller-scale, wave-2 wave activity.

The figures are clear and the paper is well-written, and is relevant to ACP. I just have one concern that would only require adding a little bit of extra discussion.

This concern has to do with the applicability of linear wave propagation and wave-mean flow interaction theory to understand the dynamics of a highly distorted vortex as it is the case. I basically refer to the ideas and results by O'Neil and Pope (1988), who argued that the separation of the flow into a slowly-evolving, zonally symmetric component and a zonal harmonics might be overreaching during the final stages of the development of an SSW. There is a clear example in Figs. 6 and 7. Lines 243-244 argue that there is wave-1 focusing on high latitudes on 20-21 Sep "guided by the poleward-displaced vortex". The authors refer to Fig 6a, where we see that the zonal-mean zonal wind on Sep 20 is poleward of 60°S and vertically aligned. However, the PV maps of Fig. 7 (or Z maps of Fig. 2) do not show a "poleward-displaced vortex", but a cyclonic vortex severely displaced off the pole and elongated. O'Neill and Pope make the case that in such situations, non-linear PV advection and vortex-vortex interactions by inspection of Ertel's PV maps might be a better suited framework to interpret the dynamics.

In this framework, it would be interesting to discuss the results of O'Neill et al (2017) on the same SSW in the Southern Hemisphere. They showed that the vortex split happened due to the interaction of a synoptic-scale cyclonic circulation in the upper troposphere barotropically aligned with one of the stratospheric vortex tips, and argued that the Eliassen-Palm fluxes cannot unequivocally be interpreted as indicating wave propagation (in this specific context of high non-zonal flows), since it is a non-local (zonally averaged) diagnostic.

We sincerely appreciate your careful review and thoughtful feedback. The comments provided were extremely helpful, and we have done our best to reflect them in the revised version to enhance the overall manuscript. In particular, in response to your major comment, we incorporated a discussion of the findings and implications of O'Neill et al. (2017) into the revised manuscript (L422–433).

**Other comments:**

- Line 121. The reversal of zonal-mean winds (..) propagated downward → progressed down to 10 hPa
   It is modified as suggested (L127).
- 2. Line 122. It is not apparent from visual inspection of Fig. 1 that the winds decelerated 100 m/s in one week. Please check.

Thank you for your comment. This question was also raised by Reviewer 2 (Comment #12). Upon reexamination, we found that the actual weakening was approximately 57 m/s. We corrected this in the revised manuscript (L128).

- 3. Line 125-126. What do the authors mean by "an upward-propagating signal from the troposphere"? The troposphere is not shown in Fig. 1, with the lower boundary at 200 hPa (i.e. lowermost stratosphere at high latitude). Besides, the polar warming seems to be confined above 100 hPa or so.
  - Thank you for your comment. This question was also raised by Reviewer 2 (Comment #14). We interpreted the upward extension of anomalously strong zonal wind deceleration (pink dots) and temperature increase (green dots) as an "upward-propagating signal," based on their evolution in the time—height cross-section (Fig. 1). However, we agree that the link between this signal and the downward propagation of easterly winds from the mesosphere is unclear. To avoid potential confusion, we removed the sentence in the revised manuscript.
- 4. Lines 204-216. I find this description of Fig. 4 not clear enough. For example: "During (...) 22–25 September, PW2 at 1 hPa predominantly exhibited westward phase speeds of up to 30 m/s". I guess a westward phase of 30m/s refers to -30m/s in the Fig. However, the amplitude of PW2 with -30m/s on 22-25 Sep is quite small, it is larger at lower phase speeds.

The original sentence was intended to indicate that PW2 at 1 hPa predominantly exhibited westward phase speeds rather than eastward ones, and that the phase speed range extended up to -30 m/s. However, as you pointed out, PW amplitudes are relatively larger at lower westward phase speeds, particularly between -25 m/s and -5 m/s. Therefore, we modified the original expression to reflect the most dominant phase speed range rather than the upper bound (L213–214).

**References:**

O'Neill, A. and Pope, V.D. (1988), Simulations of linear and nonlinear disturbances in the stratosphere. Q.J.R. Meteorol. Soc., 114: 1063-1110. https://doi.org/10.1002/qj.49711448210

O'Neill, A., Oatley, C.L., Charlton-Perez, A.J., Mitchell, D.M. and Jung, T. (2017), Vortex splitting on a planetary scale in the stratosphere by cyclogenesis on a subplanetary scale in the troposphere. Q.J.R. Meteorol. Soc., 143: 691-705. https://doi.org/10.1002/qj.2957.

**References**

- Andrews, D. G., Holton, J. R., and Leovy, C. B.: Middle Atmosphere Dynamics. Academic Press, San Diego, CA, 489 pp., ISBN 9780120585762, 1987.
- Charney, J. G. and Drazin, P. G.: Propagation of planetary-scale disturbances from the lower into the upper atmosphere, J. Geophys. Res., 66, 83–109, https://doi.org/10.1029/JZ066i001p00083, 1961.
- Jucker, M.: Scaling of Eliassen–Palm flux vectors, Atmos. Sci. Lett., 22, e1020, https://doi.org/10.1002/asl.1020, 2021.
- Kosaka, Y., Kobayashi, S., Harada, Y., Kobayashi, C., Naoe, H., Yoshimoto, K., and Onogi, K.: The JRA-3Q reanalysis, J. Meteorol. Soc. Jpn. Ser. II, 102, 49–109, https://doi.org/10.2151/jmsj.2024-004, 2024.
- Koshin, H., Yamazaki, Y. H., Tomikawa, Y., Sato, K., and Miyoshi, Y.: The JAGUAR-DAS whole neutral atmosphere reanalysis: JAWARA, Prog. Earth Planet. Sci., 12, 26, https://doi.org/10.1186/s40645-024-00674-3, 2025.
- O'Neill, A., Oatley, C. L., Charlton-Perez, A. J., Mitchell, D. M., and Jung, T.: Vortex splitting on a planetary scale in the stratosphere by cyclogenesis on a subplanetary scale in the troposphere, Q. J. R. Meteorol. Soc., 143, 691–705, https://doi.org/10.1002/qj.2957, 2017.
- Sehra, P. S.: Upper atmospheric thermal structure in Antarctica, Nature, 254, 401–404, https://doi.org/10.1038/254401a0, 1975.
- Sehra, P. S.: Antarctic atmosphere: Temperature exploration and seasonal variations, J. Geophys. Res., 81, 3715–3737, https://doi.org/10.1029/JC081i021p03715, 1976.
- Sehra, P. S.: Stratospheric circulation over Antarctica, J. Meteor. Soc. Jpn., 54, 332–340, 1979.
- Simmons, A., Hersbach, H., Dee, D., and Berrisford, P.: Global stratospheric temperature bias and other stratospheric aspects of ERA5 and ERA5.1, ECMWF Tech. Rep., https://doi.org/10.21957/rcxqfmg0, 2020.
- Song, B. G., Chun, H. Y., and Song, I. S.: Role of gravity waves in a vortex-split sudden stratospheric warming in January 2009, J. Atmos. Sci., 77, 3321–3342, https://doi.org/10.1175/JAS-D-20-0039.1, 2020.
- Vallis, G. K.: Atmospheric and Oceanic Fluid Dynamics, Cambridge University Press, Cambridge, 2017.
- WMO/IQSY: International Years of the Quiet Sun (IQSY) 1964–65. Alert messages with special references to stratwarms, WMO/IQSY Rep. No. 6, Secretariat of the World Meteorological Organization, Geneva, Switzerland, 1964.
- Yoo, J. H., Chun, H. Y., and Kang, M. J.: Vortex preconditioning of the 2021 sudden stratospheric warming: Barotropic–baroclinic instability associated with the double westerly jets, Atmos. Chem. Phys., 23, 10869–10881, https://doi.org/10.5194/acp-23-10869-2023, 2023.

---

## Referee Report (RR1)

**Review of egusphere-2025-748 version 2**

I thank the authors for their generally thorough address of the comments I made on their initial submission. There are several relatively minor comments I would still like to make, though, which I list by line number of the revised manuscript:

**19-20:** The sentence "Nonlinear wave—wave interaction drove early wave generation, while the latter played a role near the onset" is confusing. I'm not entirely sure what "the latter" refers to, and it is not clear how different "early" and "near the onset" are. Is the phrase "Nonlinear wave—wave interaction drove early wave generation" a reference to point (i) of the previous sentence, while "the latter played a role near the onset" is a reference to point (ii)? This should be rewritten.

**101-103:** This statement of the way the Charney-Stern-Pedlosky criterion is being satisfied is confusing. What you mean is "The criterion is met by  $Q_y$  changing sign within the domain" but I think the current version of the text could easily be misinterpreted as "The criterion is met by having  $Q_y$ <0 somewhere in the domain" which is not a correct statement of the criterion. The text should be updated for clarity.

**175:** "sum of two terms" should be "sum of the two terms". Doing another proofreading of the paper is probably warranted.

**201:** The listed publication year for the Jucker paper about E-P flux scaling is incorrect—it's actually 2021.

**204-206:** The discussion in the Song et al. (2020) paper is helpful, but (correct me if I'm wrong) it doesn't actually say anything about the sine and cosine tapering technique. I was not able to readily determine what was meant by this via a Google search, so some more explanation/appropriate citation to this specific signal processing technique is probably in order.

**405:** The phrase "the reason underlying the increasing frequency of SSWs with these easterlies, linked to climate change" is a little odd. Is this intended as a claim that the increasing tendency for enhanced stratopause wave driving to precede SSWs is a manifestation of radiatively forced climate change? Koushik et al. (2022) do not actually claim this, so it seems premature to make such an assertion. Alternatively if the intent is merely to suggest that there *may* exist a link to forced climate change, the language should be more tentative.

**409-412:** This claim should have a citation for it, and may be a bit exaggerated: yes, Aura MLS didn't yet exist at the time in question but as I noted in my first-round review there should have been some AMSU-A channel 14 data. Looking at the text and Table 2 of the McCarty et al. 2016 technical report cited in my previous review, that channel was likely active on NOAA-16 and NOAA-17 during SSW02.

---

## Author Response (AR2)

**Response to Reviewer #2's Comments**

**Ji-Hee Yoo and Hye-Yeong Chun**

Department of Atmospheric Sciences, Yonsei University, Seoul, Korea

**10 September 2025**

**General comment:**

I thank the authors for their generally thorough address of the comments I made on their initial submission. There are several relatively minor comments I would still like to make, though, which I list by line number of the revised manuscript:

We thank the reviewer for the careful second review. Your feedback has substantially improved the manuscript. We have addressed all remaining comments in detail below and revised the text accordingly.

**Minor Comments:**

1. 19-20: The sentence "Nonlinear wave—wave interaction drove early wave generation, while the latter played a role near the onset" is confusing. I'm not entirely sure what "the latter" refers to, and it is not clear how different "early" and "near the onset" are. Is the phrase "Nonlinear wave—wave interaction drove early wave generation" a reference to point (i) of the previous sentence, while "the latter played a role near the onset" is a reference to point (ii)? This should be rewritten.

In the original text, "nonlinear wave—wave interaction" corresponds to mechanism (i) and "the latter" was intended to refer to mechanism (ii) (over-reflection of upward-propagating PW2). By "early" we meant the first two days of the 4-day wave-generation period (from Lag = -3 to Lag = -2), and by "near the onset" we meant the final two days (from Lag = -1 to Lag = 0). To avoid any confusion, we stated the mechanisms and timing more explicitly in the revised manuscript (L18–20).

2. 101-103: This statement of the way the Charney-Stern-Pedlosky criterion is being satisfied is confusing. What you mean is "The criterion is met by Qy changing sign within the domain" but I think the current version of the text could easily be misinterpreted as "The criterion is met by having Qy<0 somewhere in the domain" which is not a correct statement of the criterion. The text should be updated for clarity.

We agree that the original sentence could be misread as the criterion being satisfied by  $\bar{q}_y < 0$  somewhere in the domain, rather than by a sign reversal of  $\bar{q}_y$  within the domain. We revised the text to state this explicitly (L101–104).

3. 175: "sum of two terms" should be "sum of the two terms". Doing another proofreading of the paper is probably warranted.

Thank you for pointing this out. We changed the phrase to "sum of the two terms" (L176) and have proofread the manuscript again and made minor editorial corrections. We will also carefully review the proofs at the production stage.

4. 201: The listed publication year for the Jucker paper about E-P flux scaling is incorrect—it's actually 2021.

Thank you for the correction. We corrected the citation year to 2021 for Jucker (L202).

5. 204-206: The discussion in the Song et al. (2020) paper is helpful, but (correct me if I'm wrong) it doesn't actually say anything about the sine and cosine tapering technique. I was not able to readily determine what was meant by this via a Google search, so some more explanation/appropriate citation to this specific signal processing technique is probably in order.

Thank you for the helpful suggestion. You are correct that Song et al. (2020) did not apply a sine/cosine taper. In our initial tests, Fourier decomposition without tapering exhibited spectral leakage, so we applied a symmetric 3-day raised-cosine (Tukey) taper to each 11-day window prior to decomposition in this study. We added a more detailed explanation and citation for this windowing technique in the figure caption.

6. 405: The phrase "the reason underlying the increasing frequency of SSWs with these easterlies, linked to climate change" is a little odd. Is this intended as a claim that the increasing tendency for enhanced stratopause wave driving to precede SSWs is a manifestation of radiatively forced climate change? Koushik et al. (2022) do not actually claim this, so it seems premature to make such an assertion. Alternatively if the intent is merely to suggest that there may exist a link to forced climate change, the language should be more tentative.

Thank you for this important point. As the reviewer notes, Koushik et al. (2022) reported an increase in the fraction of SSWs preceded by equatorial upper-stratospheric easterlies after 2000, but did not attribute this trend to climate change. We revised the sentence to use clearly tentative language (L406–L407).

7. 409-412: This claim should have a citation for it, and may be a bit exaggerated: yes, Aura MLS didn't yet exist at the time in question but as I noted in my first-round review there should have been some AMSU-A channel 14 data. Looking at the text and Table 2 of the McCarty et al. 2016 technical report cited in my previous review, that channel was likely active on NOAA-16 and NOAA-17 during SSW02.

Thank you for the careful reading. You are right that our original sentence overstated the lack of observations by omitting the AMSU-A channel-14 radiances assimilated in MERRA-2 during 2002 (McCarty et al., 2016). We revised the sentence to acknowledge this constraint and added the appropriate citation. We now state that AMSU-A channel-14 provides sensitivity mainly in the upper stratosphere (~30–45 km), so the pre-2004 upper stratosphere/lower mesosphere remains only weakly constrained (L412–415).

**References**

Jucker, M.: Scaling of Eliassen–Palm flux vectors, Atmos. Sci. Lett., 22, e1020, https://doi.org/10.1002/asl.1020, 2021.

Koushik, N., Kumar, K. K., and Pramitha, M.: A tropical stratopause precursor for sudden stratospheric warmings, Sci. Rep., 12, 2937, https://doi.org/10.1038/s41598-022-06864-7, 2022.

McCarty, W., Coy, L., Gelaro, R., Huang, A., Merkova, D., Smith, E. B., Sienkiewicz, M., and Wargan, K.: MERRA-2 input observations: Summary and assessment, NASA Technical Report Series on Global Modeling and Data Assimilation, NASA/TM–2016–104606/Vol. 46, NASA Goddard Space Flight Center, Greenbelt, MD, USA, 64 pp., 2016. available at: https://ntrs.nasa.gov/citations/20160014544 (last access: 10 September 2025).

Song, B. G., Chun, H. Y., and Song, I. S.: Role of gravity waves in a vortex-split sudden stratospheric warming in January 2009, J. Atmos. Sci., 77, 3321–3342, https://doi.org/10.1175/JAS-D-20-0039.1, 2020.